# Epigenetic centromere identity is precisely maintained through DNA replication but is uniquely specified among human cells

Megan A Mahlke[1,2], Lior Lumerman[1,2], Peter Ly[3], Yael Nechemia-Arbely[1,2]

Centromere identity is defined and maintained epigenetically by the presence of the histone variant CENP-A. How centromeric CENP-A position is specified and precisely maintained through DNA replication is not fully understood. The recently released Telomere-to-Telomere (T2T) genome assembly containing the first complete human centromere sequences provides a new resource for examining CENP-A position. Mapping CENP-A position in clones of the same cell line to the T2T assembly identified highly similar CENP-A position after multiple cell divisions. In contrast, centromeric CENP-A epialleles were evident at several centromeres of different human cell lines, demonstrating the location of CENP-A enrichment and the site of kinetochore recruitment vary among human cells. Across the cell cycle, CENP-A molecules deposited in G1 phase are maintained in their precise position through DNA replication. Thus, despite CENP-A dilution during DNA replication, CENP-A is precisely reloaded onto the same sequences within the daughter centromeres, maintaining unique centromere identity among human cells.

## Introduction

Centromere position is specified epigenetically by the centromeric histone H3 variant CENP-A (Cleveland et al, 2003; Barnhart et al, 2011; Black & Cleveland, 2011; Fachinetti et al, 2013). The precise deposition and maintenance of CENP-A across the cell cycle is the initial step in establishing functional centromeres, which are essential for assembling the kinetochore to allow for faithful chromosome segregation during mitosis, thereby preserving genomic integrity. CENP-A deposition in early G1 is tightly regulated by the combined activity of the Mis18 licensing complex (Hayashi et al, 2004; Fujita et al, 2007; Nardi et al, 2016; Stellfox et al, 2016; Pan et al, 2017; Spiller et al, 2017) and CENP-A's chaperone HJURP (Dunleavy

et al, 2009; Foltz et al, 2009; Barnhart et al, 2011; Zasadzińska et al, 2013; Wang et al, 2014; Tachiwana et al, 2015; Pan et al, 2019). During subsequent DNA replication, HJURP (Zasadzińska et al, 2013), CENP-C, and the CCAN (Nechemia-Arbely et al, 2019) are required for the retention and reloading of CENP-A at centromeres of daughter DNA strands. CENP-A distribution between the two daughter centromeres during DNA replication results in dilution of CENP-A to ~50% occupancy until the following G1 (Jansen et al, 2007; Nechemia-Arbely et al, 2012; Silva et al, 2012; Stankovic et al, 2017). The temporal separation between CENP-A dilution in S phase and new CENP-A deposition in G1 raises the important question of how centromere epigenetic identity is maintained across the cell cycle (Mahlke & Nechemia-Arbely, 2020).

Human centromeres are megabase-long chromosomal regions (Wevrick & Willard, 1989) composed of tandemly repeated 171 base pair (bp) α-satellite monomers organized into high-order repeat (HOR) arrays (Willard, 1985; Willard & Waye, 1987) that cannot be resolved with traditional sequencing approaches (Eichler et al, 2004; Miga, 2015). Thus, centromeres in the hg38 human genome assembly (GRCh38p.13) are represented by models that reflect the observed variation in human α-satellite repeat sequences but do not distinguish between centromeric and pericentromeric sequences, and arbitrarily assign the order of repeats in each centromeric array (Levy et al, 2007; Miga et al, 2014; Schneider et al, 2017). Moreover, attempts to use sequencing-based approaches to study centromere epigenetics are complicated by the inability to accurately map short sequencing reads to a precise centromeric location or to interpret whether mapped sequencing reads represent the true position of epigenetic modifications in centromeric chromatin. Consequently, the highly complex and repetitive nature of human centromere sequences hinders the study of centromere genomics and epigenetics, limiting our understanding of CENP-A maintenance.

The complete CHM13 human genome assembly released by the Telomere-to-Telomere (T2T) Consortium represents a significant achievement that uncovers the true sequences of large and repetitive genomic elements using long-read sequencing technologies

[1]UPMC Hillman Cancer Center, Pittsburgh, PA, USA   [2]Department of Pharmacology and Chemical Biology, University of Pittsburgh, Pittsburgh, PA, USA   [3]Department of Pathology, University of Texas Southwestern Medical Center, Dallas, TX, USA

Correspondence: arbelyy@upmc.edu

(Nurk et al, 2022), including 156.2 Mb of centromeric satellites that remained missing from the hg38 assembly (GRCh38p.13), 21 years after the first human genome was released (Lander et al, 2001). Here, we used the T2T assembly (CHM13v1.1) that contains the first accurate description of human centromere sequences to assess the pattern of centromeric CENP-A binding in various human cell types and across the cell cycle. Using new CENP-A ChIP-seq datasets, as well as our own (Nechemia-Arbely et al, 2019) and other publicly available CENP-A ChIP-seq and CUT&RUN data (Thakur & Henikoff, 2016; Dumont et al, 2020; Logsdon et al, 2021), we demonstrate that CENP-A position varies significantly within the same HOR between different human cell lines, indicating the presence of CENP-A epialleles at several human centromeres. In contrast to the plasticity in CENP-A position between different cell types, in the same cell line, CENP-A position is maintained within the HOR from parental cells to derived clones through multiple cell divisions. Furthermore, we test the sequences bound by CENP-A in G1- and G2-enriched cells and demonstrate that CENP-A is maintained at the same $\alpha$-satellite sequences by precise reloading at the same position during DNA replication, thereby preserving centromere epigenetic identity.

# Results

## The position and distribution of CENP-A at human centromeres differs significantly between hg38 and T2T genome assemblies

Though the position of $\alpha$-satellite monomer repeats in the centromere models contained within the hg38 assembly (GRCh38p.13) are arbitrary and do not represent true centromere HOR arrangement, the hg38 centromere models have been used to estimate the distribution of CENP-A at human centromeres (Nechemia-Arbely et al, 2019; Hoffmann et al, 2020). To investigate the true position of CENP-A at human centromeres using the recently released T2T genome assembly (CHM13v1.1) (Nurk et al, 2022), we performed CENP-A ChIP-seq in PD-NC4 fibroblasts (Amor et al, 2004). To obtain a sufficient number of cells for CENP-A ChIP and to derive single-cell clones, we first immortalized and partially transformed PD-NC4 cells by expressing *hTERT* and oncogenic *KRAS^V12*, respectively. We mapped the resulting data (Table S1) alongside previously generated CENP-A ChIP-seq data from HeLa cells (Nechemia-Arbely et al, 2019) to the T2T (CHM13v1.1) and hg38 (GRCh38p.13) genome assemblies, retaining all high-quality centromeric reads and assigning multi-mapping repetitive reads randomly to a single location (Fig 1A, top panel).

Mapping of CENP-A–bound DNA reads revealed dramatic differences in CENP-A position at nearly all centromeres between the hg38 and T2T genome assemblies (Figs 1B–D and S1). Some differences in CENP-A mapping between the two assemblies were the result of uncovering the actual arrangement of HORs and other repetitive elements at centromeres in the T2T assembly. At chromosomes 1 (Fig 1B), 6, 19, and 22 (Fig S1), the T2T assembly contains a longer span of centromeric HOR sequences than was predicted in the hg38 centromere model. At chromosomes 3 (Fig 1D) and 4 (Fig S1), the T2T assembly includes large insertions of human satellite (Hsat) sequences in the center of the HOR array (Altemose et al,

2022), splitting CENP-A into two separate centromeric regions and dramatically changing the centromeric CENP-A landscape between the hg38 and T2T assemblies. At other centromeres, we discovered distinctive CENP-A enrichment patterns between the two assemblies that were not clearly related to changes in HOR size or organization. At the centromere of chromosome 2, CENP-A is distributed uniformly across the HOR in both HeLa and PD-NC4 cells aligned to the hg38 assembly but is locally enriched in distinct centromeric regions in HeLa and PD-NC4 cells when aligned to the T2T assembly (Fig 1C). These cell line–specific regions of CENP-A enrichment were also evident at the centromeres of other chromosomes (see chromosomes 6, 8, 9, 11, 17, 18, 22, and X in Fig S1). Alignment to the T2T assembly also highlighted the small-scale differences in CENP-A position between HeLa and PD-NC4 cells (Fig 1E–G). These striking differences in CENP-A position between HeLa and PD-NC4 cells aligned to the T2T assembly suggest that the centromeric site enriched for CENP-A binding may be specified at different positions within the same centromeres in different human cells.

## CENP-A position varies between human cell lines

Recent reports have identified that human centromeres evolve through layered expansions leading to generation of distinct sets of $\alpha$-satellite repeats within the same HOR that can be clustered based on their shared sequence variants into higher order repeat haplotypes or "HOR-haps" (Altemose et al, 2022). Furthermore, CENP-A enrichment and the site of kinetochore recruitment in CHM13 cells were reported to frequently occur at the evolutionarily younger HOR-hap and the site of recent expansion (Altemose et al, 2022). Despite this progress in understanding centromere evolution and its effect on centromere structure, the location of centromeric CENP-A enrichment at all centromeres among diverse human cells remains largely unexplored.

By mapping CENP-A ChIP-sequencing data from CHM13, PD-NC4, and HeLa cells to the T2T assembly and using the recent characterization of the active HOR S3CXH1L (DXZ1 in hg38) within the X centromere into distinct HOR-haps (Altemose et al, 2022), we found that CENP-A in PD-NC4 and CHM13 cells is enriched at the evolutionarily younger HOR-hap and the site of recent HOR expansion within the S3CXH1L HOR, whereas CENP-A in HeLa cells is enriched at an evolutionarily older HOR-hap of S3CXH1L that does not overlap a site of recent expansion (Fig 2A). Notably, CENP-A enrichment at the X centromere in HeLa cells occupies a larger span (1.2 Mb, Fig 2A) that is roughly equivalent to the length of two active centromeres in PD-NC4 or CHM13, suggesting that the active centromere occupies two adjacent regions in HeLa maternal and paternal X centromeres. In contrast, examination of the active HOR of chromosome 2, S2C2H1L (D2Z1 in hg38), revealed that CENP-A in both CHM13 and HeLa cells is enriched at an evolutionarily older HOR-hap and site of recent HOR expansion, whereas in PD-NC4 cells, CENP-A is enriched at an evolutionarily younger HOR-hap that also overlaps a site of recent expansion (Fig 2B). Previous work has shown that two $\alpha$-satellite HOR arrays within the centromere of chromosome 17, S317H1L (D17Z1) and S317H1-B (D17Z1-B), are both capable of recruiting CENP-A and forming a kinetochore in two fibroblast cell lines (FIBL1-HCT116 and FIBL4-HT1080D) (Maloney

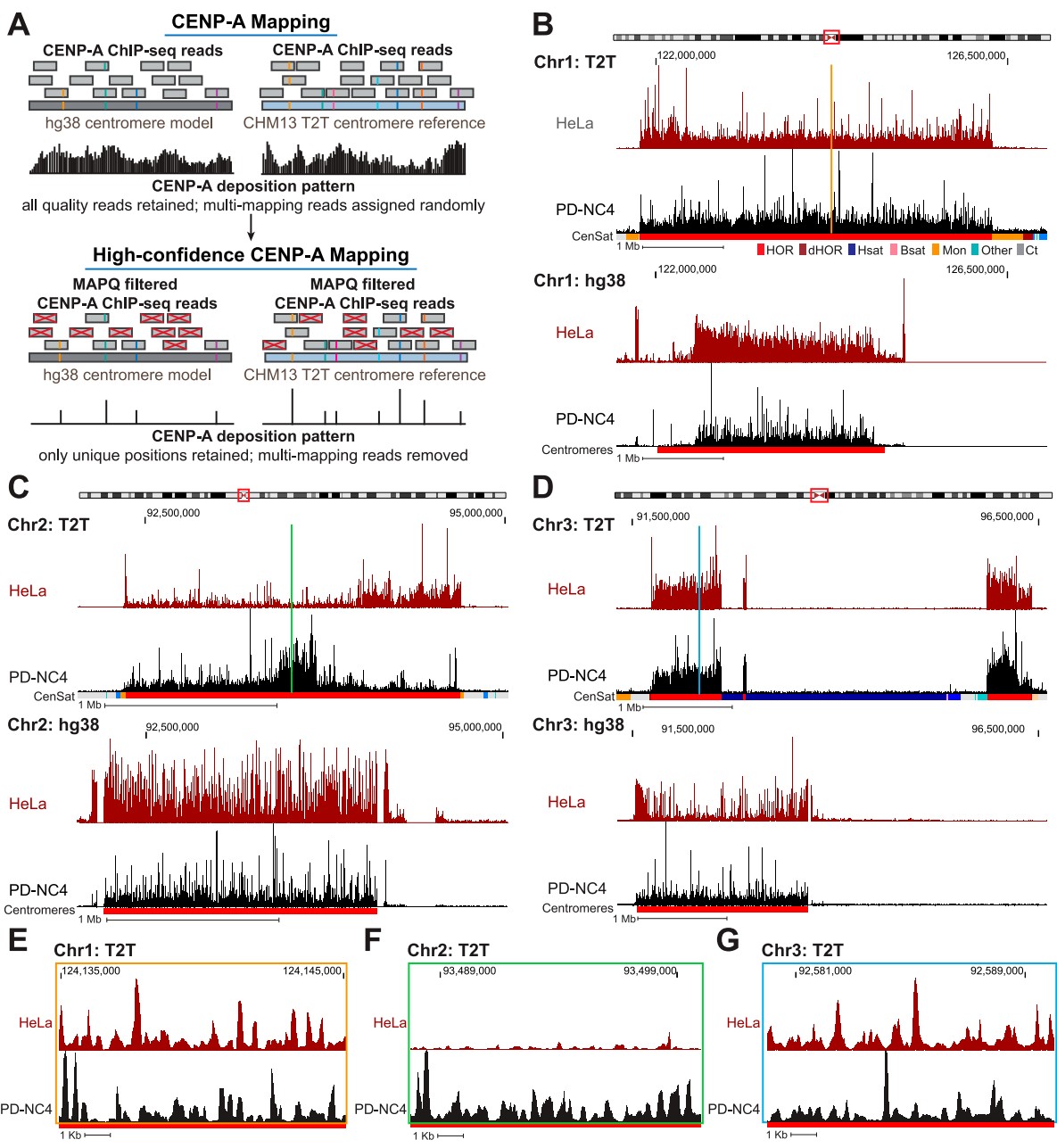

**Figure 1. CENP-A alignment patterns differ dramatically between hg38 centromere reference models and T2T centromere sequences.**
**(A)** Schematic representing conventional mapping of CENP-A–bound reads (top panel) and MAPQ-filtered high-confidence mapping of CENP-A–bound reads (bottom panel) to the hg38 (left) and T2T (right) assemblies. The higher number of unique variants in the T2T assembly allows for a greater number of high-confidence CENP-A reads to be mapped. Reads with colored lines represent unique variants within the α-satellite repeats. **(B, C, D)** Position of CENP-A reads in HeLa (maroon) and PD-NC4 (black) cells when mapped to the T2T (top) and hg38 (bottom) assemblies at the centromeres of chromosomes 1 (B), 2 (C), and 3 (D). Chromosome ideograms at top indicate position of the centromere. Annotation tracks below the T2T assembly indicate positions of centromeric satellites; see the color key in panel (B). (HOR = higher order repeats, red; dHOR = divergent higher order repeats, maroon; Hsat = human satellites, blue; Bsat = beta satellites, pink; Mon = monomers, orange; Others = other centromeric satellites, teal; Ct = centric satellite transition regions, grey). Annotation tracks below the hg38 assembly indicate positions of centromeres (red). Scale bar, 1 Mb. **(E, F, G)** High-resolution view of CENP-A reads in HeLa (maroon) and PD-NC4 (black) cells when mapped to the T2T assembly at the same centromeres shown in (B, C, D). Color-coded panels represent the colored locations indicated in (B, C, D). Scale bar, 1 Kb.

et al, 2012). In the cell lines we examined, CENP-A was only found at the S317H1L (D17Z1) HOR but was localized at different HOR-haps within the array (Fig 2C). In CHM13 cells, CENP-A is enriched at the evolutionarily younger HOR-hap and is highly enriched at the zone of recent expansion. In PD-NC4 cells, CENP-A is enriched at an evolutionarily older HOR-hap flanking the right side of the younger HOR-hap (at T2T coordinates of ~27 Mb). In HeLa cells, CENP-A is enriched at an evolutionarily older HOR-hap that flanks the

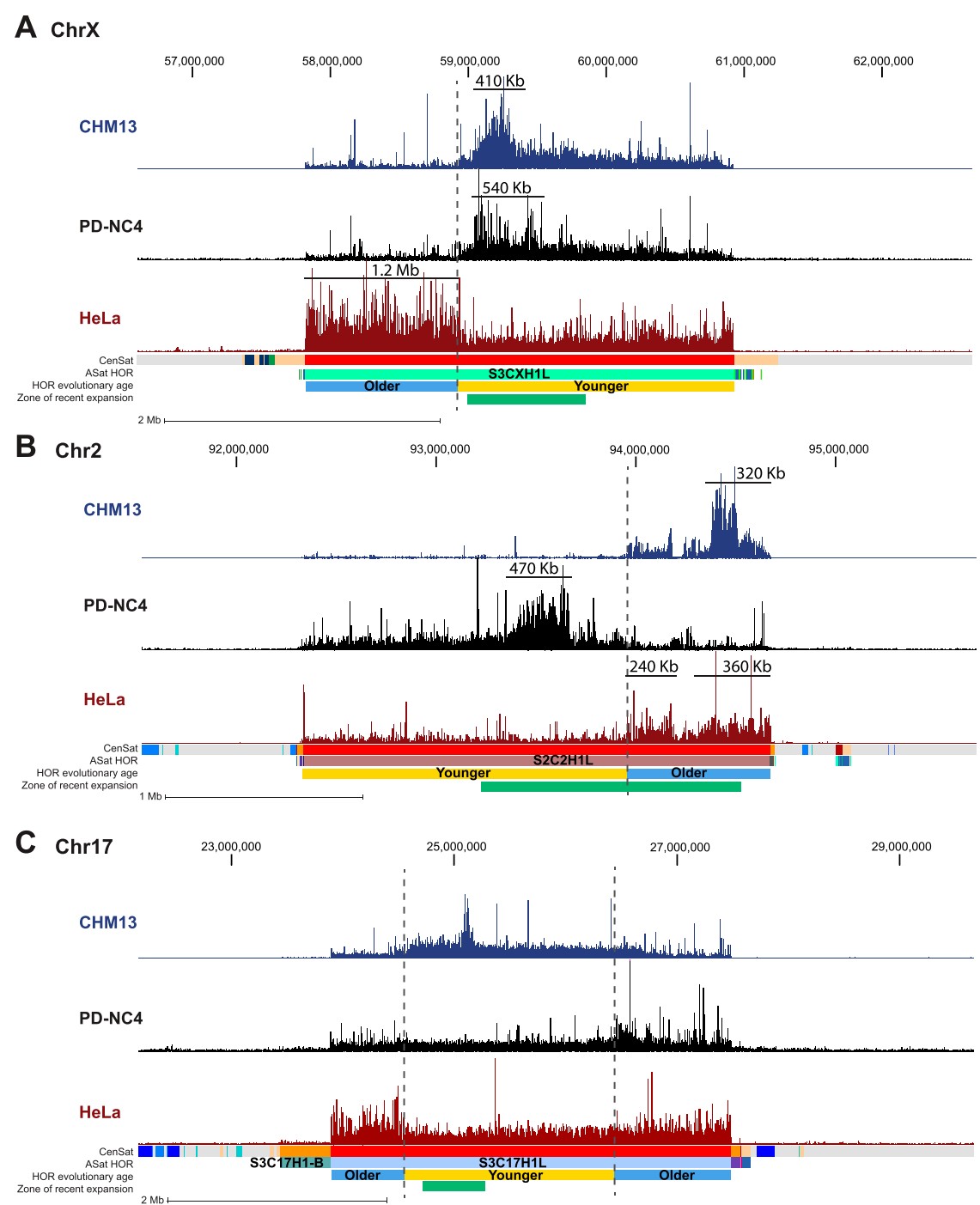

**Figure 2. CENP-A binds at evolutionarily younger and older HOR-haps in different centromeres and different cell lines.**
**(A, B, C)** Mapping of CENP-A-bound reads in CHM13 (blue), PD-NC4 (black), and HeLa cells (maroon) at the centromeres of chromosome X (A), chromosome 2 (B), and chromosome 17 (C). Dashed lines mark regions that are differentially enriched for CENP-A binding between cell lines. Annotation tracks indicate positions of centromeric satellites (CenSat), higher order repeat arrays (ASat HOR), rough HOR age based on evolutionary age of HOR-haps (HOR evolutionary age), and the location of the zone of recent HOR expansion (zone of recent expansion). HOR age and zone of recent expansion were adapted from (Altemose et al, 2022). Black lines in (A, B) indicate the active centromere size in Kb.

opposite side of the younger HOR-hap (at T2T coordinates of ~24 Mb). Interestingly, HeLa cells also exhibit a lower level of CENP-A enrichment at the evolutionarily older HOR-hap flanking the right side of the younger HOR-hap (at T2T coordinates of ~27 Mb), which

may indicate that HeLa cells have distinct maternal and paternal CENP-A epialleles on opposite ends of the chromosome 17 centromere (Fig 2C). Taken together, these distinct patterns of centromeric CENP-A enrichment in different human cell lines suggest

the presence of previously undocumented CENP-A epialleles (i.e., different positioning of the region enriched for CENP-A binding with respect to the underlying α-satellite DNA) within a single human centromeric HOR.

To further investigate centromeric CENP-A epialleles in human cell lines, we compared CENP-A position in HeLa (cervical cancer), PD-NC4 (fibroblast), RPE-1 (retinal pigment epithelium), and HuRef (lymphoblastoid cell line [Levy et al, 2007]) cells when aligned to the T2T assembly, using new (PD-NC4) and previously published CENP-A ChIP-seq ([Nechemia-Arbely et al, 2019] for HeLa; [Logsdon et al, 2021] for CHM13; [Henikoff et al, 2015] for HuRef) and CUT&RUN ([Dumont et al, 2020] for RPE-1) datasets. We found that CENP-A position varies between human cell lines in the centromeres of almost all chromosomes (excluding chromosomes 4, 15, 16, and 21, at which CENP-A pattern is similar across cell lines) when mapped to the T2T genome assembly (Figs 3A–C and S2). Conversely, positions of centromeric CENP-A are highly similar in PD-NC4 parental cells and two PD-NC4 single-cell–derived clones after ~40–50 cell divisions required to isolate clones and generate enough cells for ChIP-seq (Fig 3G), indicating that within the same cell line CENP-A position is maintained over many cellular divisions and that the observed CENP-A epialleles between human cell lines of different origin truly represent individual or cell type–specific changes in CENP-A position. At some centromeres, a single region enriched for CENP-A binding, usually ranging between 200–500 kb (as has previously been documented for the CENP-A–enriched region [Altemose et al, 2022]), was easily identified, although the location of the region enriched for CENP-A may differ between cell lines (in Figs 2A and B, 3A and C, and S2, see centromeres of chromosomes 2, 6, 8, 10, 12, 14, 16, 17, 18, and 20). At other centromeres, a larger contiguous region or two separate regions enriched for CENP-A binding were evident, suggesting differential CENP-A position between maternal and paternal centromeres/chromosomes in diploid cell lines (Fig 2A and B, HeLa cells; Fig 3A, RPE-1 cells; Fig 3C, PD-NC4 cells). However, a definitive region enriched for CENP-A was challenging to identify in some centromeres of different cell lines (in Figs 3B and S2, e.g., see the HeLa and PD-NC4 centromeres of chromosomes 3 and 4, and all lines in chromosome 21).

Recognizing that conventional mapping of CENP-A–bound DNAs can generate a bias because of random assignment of multi-mapping reads, we produced high-confidence centromeric CENP-A alignments, which can be achieved with two methods: (1) filtering each dataset to retain only high-confidence CENP-A alignments based on MAPQ score (MAPQ>20), effectively removing any reads that map to more than one location and retaining only the highest confidence alignments at unique centromeric sequences (Fig 1A, bottom panel), or (2) K-mer assisted mapping that identifies unique k-mers within centromeric HORs and retains reads mapping to these locations only (Altemose et al, 2022). Mapping of high-confidence CENP-A reads using MAPQ and k-mer approaches was highly concordant (Fig S3). We therefore chose to continue with the MAPQ approach, which is consistent with our previous methodology (Nechemia-Arbely et al, 2019). Though retaining only the highest confidence reads prevents the obfuscation of CENP-A position because of random mapping, it also removes ~90–95% of centromeric CENP-A reads because of the highly repetitive nature of centromeric HOR arrays, limiting the analysis to only 5–10%

of CENP-A–bound DNAs (Fig 1A and Table S2). Of note, the RPE-1 CUT&RUN dataset retained a much higher percentage (24%) of reads after MAPQ filtering compared with HeLa, PD-NC4, and HuRef ChIP-seq datasets (Table S2). We found that high-confidence CENP-A positions also differ between human cell lines at most centromeres (Figs 3D–F and S2), supporting our findings using conventional high-quality CENP-A mapping and validating the existence of previously undocumented CENP-A epialleles within a single centromeric HOR at several human centromeres. However, retaining only high-confidence CENP-A reads reduced the apparent variability in CENP-A deposition patterns between human cell lines, as evidenced by PCA analysis (Fig 3H), highlighting the importance of both conventional high-quality mapping and high-confidence mapping strategies when evaluating CENP-A position and distribution at centromeres. Taken together, our analyses suggest that variability in CENP-A placement may be a natural feature of centromeres in different human cell lines.

## CENP-A position is maintained through DNA replication with precision

Using CENP-A ChIP-sequencing from HeLa cells enriched in G1 and G2 and mapped to the centromere reference models within the hg38 assembly, we previously showed that CENP-A is precisely retained at the same centromeric sequences through DNA replication (Nechemia-Arbely et al, 2019). Because our current analysis reveals that mapping of the same data to both hg38 and T2T genome assemblies yields significantly different CENP-A distribution patterns (Fig 1B–G), we further tested our hypothesis of precise CENP-A retention through DNA replication by mapping our previous CENP-A ChIP-sequencing data from HeLa cells enriched in G1 and G2 (Nechemia-Arbely et al, 2019) to the T2T genome assembly. Centromeric CENP-A distribution in cells enriched in the G1 phase was highly similar to its distribution in cells enriched in the G2 phase at all human centromeres (Figs 4A, B, E, and F and S4A). Almost all (~93%) significantly enriched CENP-A peaks at α-satellite sequences (MACS2 P < 0.00001, ≥10-fold enrichment; see the Materials and Methods section) present during G1 remained at the same sequences in G2 (Fig 4I). Though CENP-A position was highly similar, it was not identical between the G1 and G2 phases, suggesting the possibility of low-level CENP-A dynamics within the population. However, most CENP-A peaks are maintained from the G1 to the G2 phase, demonstrating CENP-A retention at the same centromeric sequences from its initial loading at early G1 through DNA replication.

Across all centromeres in the hg38 assembly, high-confidence mapping previously identified 96 unique single-copy CENP-A binding sites in HeLa cells (Nechemia-Arbely et al, 2019). Using the same HeLa dataset aligned to the T2T assembly, we identified 1,879 unique single-copy CENP-A binding sites in HeLa cells that are significantly enriched for CENP-A (MACS2 P < 0.00001, ≥10-fold enrichment). Within this 20-fold expanded set of unique high-confidence CENP-A binding sites, ~97% of G2 CENP-A peaks were maintained at their G1 positions, demonstrating precise reloading of CENP-A at the same position during DNA replication (Fig 4C, G, and J). High-resolution nucleosomal views demonstrate retention of CENP-A nucleosomes at a population level through DNA

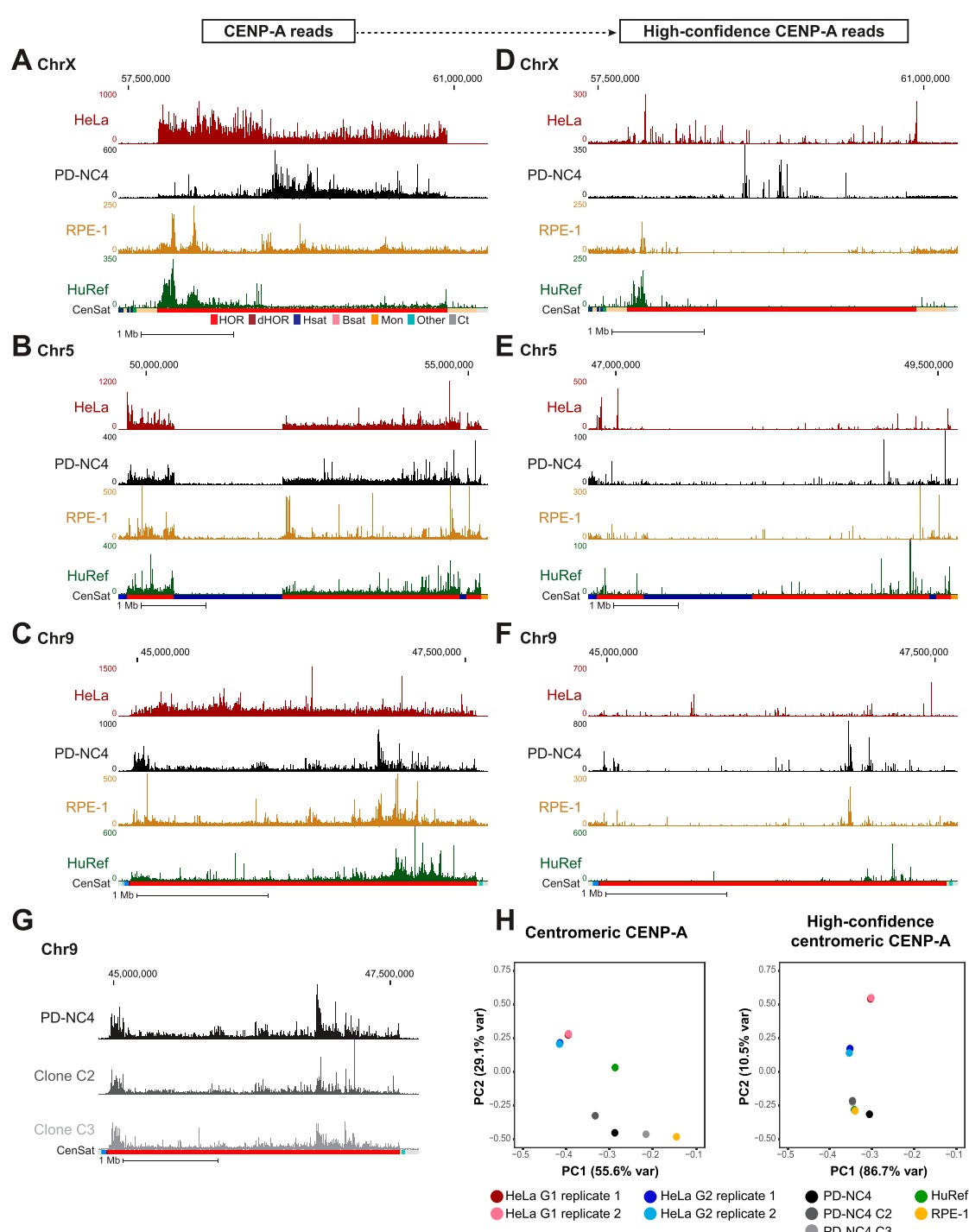

**Figure 3.  CENP-A position and high-confidence CENP-A binding sites differ between human cell lines.**
**(A, B, C)** Mapping of CENP-A to chromosome X (A), chromosome 5 (B), and chromosome 9 (C) of the T2T assembly in HeLa (maroon), PD-NC4 (black), RPE-1 (yellow), and HuRef (green) cells. Scale bar, 1 Mb. Annotation tracks below the T2T assembly indicate positions of centromeric satellites; see color key in panel (A): (HOR = higher order repeats, red; dHOR = divergent higher order repeats, maroon; Hsat = human satellites, blue; Bsat = beta satellites, pink; Mon = monomers, orange; Others = other centromeric satellites, teal; Ct = centric satellite transition regions, grey). **(D, E, F)** Mapping of high-confidence CENP-A to chromosome X (D), chromosome 5 (E), and chromosome 9 (F) of the T2T assembly in HeLa, PD-NC4, RPE-1, and HuRef cells. Scale bar, 1 Mb. **(G)** Mapping of CENP-A reads from parental PD-NC4 fibroblasts and two single-cell–derived clones to the centromere of chromosome 9 in the T2T assembly. **(H)** PCA plots of significantly enriched centromeric CENP-A peaks (MACS2, *P* < 0.00001) at all (left panel) and high-confidence (right panel) positions in different human cell lines. HeLa G1 replicates, maroon and pink; HeLa G2 replicates, dark and light blue; PD-NC4 parental and derived clones, black and grey; RPE1, yellow; HuRef, green.

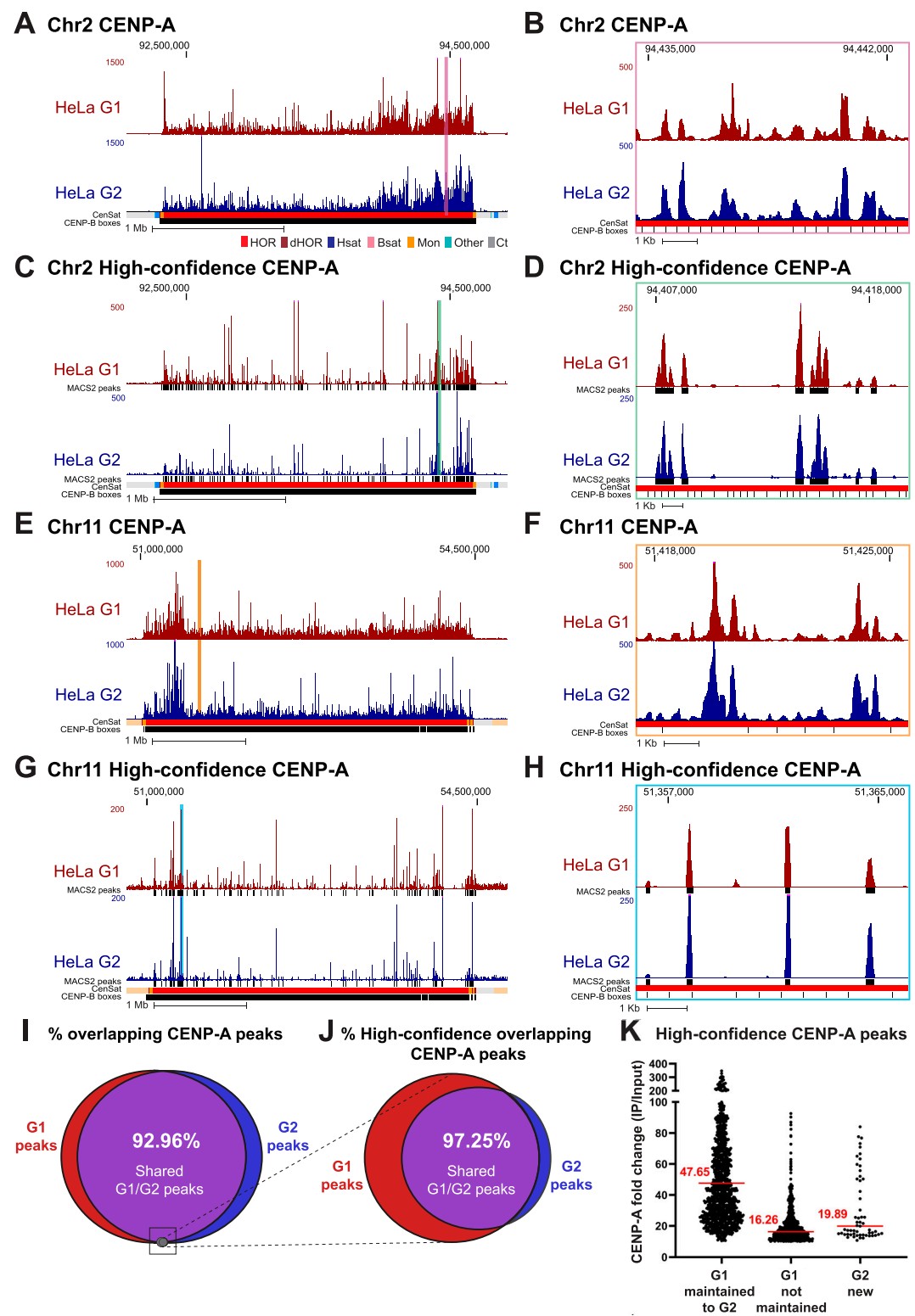

**Figure 4. CENP-A nucleosomes are retained through DNA replication at human centromeres.**
**(A, B, C, D)** CENP-A position in HeLa cells enriched at G1 (maroon) and G2 (blue) mapped to chromosome 2 of the T2T assembly with conventional mapping (A, B) and high-confidence mapping (C, D). (A, C); scale bar 1 Mb. (B, D); scale bar 1 Kb. Color coded panels (B, D) represent high-resolution view of the colored locations indicated in (A, C). MACS2 peaks in (C, D) in black indicate positions significantly enriched peaks for CENP-A (MACS2 P < 0.00001). Annotation tracks below the T2T assembly indicate positions of centromeric satellites; see color key in panel (A): (HOR = higher order repeats, red; dHOR = divergent higher order repeats, maroon; Hsat = human satellites, blue; Bsat = beta satellites, pink; Mon = monomers, orange; Others = other centromeric satellites, teal; Ct = centric satellite transition regions, grey). **(E, F, G, H)** CENP-A

replication at the same sequence and centromeric position with precision (Figs 4D and H and S4B). 33.7% of high-confidence CENP-A G1 peaks were not maintained from G1 to G2 (Fig 4J, red). However, these high-confidence lost G1 peaks were ~2.9-fold less enriched for CENP-A binding compared with G1 peaks that were maintained in G2 (Figs 4K and S5C), indicating that lost G1 peaks were not a hot spot for CENP-A deposition at G1 within the cell population and represent sites that are less likely to be occupied by CENP-A. Moreover, because the MAPQ-filtered dataset represents only 5.63% of total CENP-A–bound DNAs in HeLa (Table S2) and 5.1% of significantly enriched CENP-A peaks (Fig 4I, grey circle), the lost G1 peaks represent only 1.89% of total CENP-A–bound DNAs in G1, whereas most G1 peaks (~93%) are retained through DNA replication (Fig 4I). A small subset of new CENP-A peaks was detected in G2 (Fig 4J, blue). Similar to the lost G1 peaks, the new G2 peaks were enriched ~2.4-fold less for CENP-A binding than peaks that were maintained from G1 to G2, indicating that CENP-A rarely occupies new positions in G2 on a population level. Taken together, most of the centromeric CENP-A–enriched peaks on a population level were maintained at the same sequences and at the same position through DNA replication, demonstrating a strong preference on the population level for precise retention of centromeric CENP-A nucleosomes during DNA replication.

# Discussion

The first assembly of complete human centromere sequences as part of an entire T2T human genome (CHM13v1.1) increases the accuracy of mapping CENP-A nucleosomes within these highly complex genomic loci. True centromere sequences remain highly challenging for short read mapping approaches, as short reads often lack unique sequences that can be used to confidently anchor their position. However, the T2T assembly of human centromere sequences vastly increases the number of known unique centromeric variants that can be used to confidently anchor the positions of centromeric proteins (Aganezov et al, 2022).

Using conventional and high-confidence mapping approaches, we demonstrate that CENP-A position at centromeres varies significantly between human cell lines mapped to the T2T assembly (Figs 1–3). This indicates that different human cell lines have distinct CENP-A epialleles at several centromeres and that the site of CENP-A enrichment, which recruits the kinetochore, is found at different locations within centromeric α-satellite DNA. Indeed, CENP-A epialleles have been documented on different HORs within the centromere of human chromosome 17 (Maloney et al, 2012) and

chromosome 7 (McNulty et al, 2017) and at different locations within a single HOR in human chromosome X (Altemose et al, 2022). CENP-A epialleles have also been documented in inbred mouse strains (Arora et al, 2022 Preprint). Here, we find that CENP-A epialleles occur at different locations within the same HOR array of a centromere and may correlate with distinct centromeric HOR-haps. In contrast, within the same cell line, the centromeric CENP-A position is maintained from parental cells to single-cell–derived clones, demonstrating faithful CENP-A maintenance across many cell divisions. The underlying cause for the differences in CENP-A position we observed in this diverse panel of human cell types remains to be determined.

One explanation for differences in CENP-A position could be innate genomic variability in centromere HOR array size and/or organization at the individual and population level (Miga, 2019) which is likely to differ between human cell lines of distinct origins. Thus, enrichment of CENP-A at different positions in human cell lines could reflect over-representation of a portion of CENP-A–bound reads in a sample because of genomic expansion of a particular HOR, or HOR-hap, relative to the T2T assembly. CENP-A position within the active HOR can also differ between maternal and paternal chromosomes in normal diploid cell lines (Maloney et al, 2012; Miga, 2019), such as the lines examined in this study, which may further complicate our interpretation of CENP-A position when aligned to the functionally haploid CHM13-derived T2T centromere sequences (Altemose et al, 2022). Alternatively, the possibility that CENP-A position at centromeres changes during the development of different cell types (fibroblast, lymphocyte, etc.) within an individual has yet to be investigated and could also contribute to the distinct localization of CENP-A within this diverse panel of human cell types.

Our results also reveal that it can be difficult to identify a single region of CENP-A enrichment within the centromere HOR when aligning non-CHM13 cell lines to the T2T assembly in many of the centromeres (Fig S1). Previous studies of CENP-A position using the T2T assembly have primarily used the CHM13 cell line, as this approach minimizes complications that may arise from mapping data generated from other human cell lines to the CHM13-derived T2T assembly. However, until individualized telomere-to-telomere reference genomes are produced for multiple diploid human cell lines such as those examined herein, the CHM13-derived T2T assembly is the best tool for mapping to centromeres and other previously unresolved repetitive regions. Our findings highlight the intriguing flexibility in CENP-A enrichment on human α-satellite arrays and underscore the need for T2T genome assemblies from additional individuals to capture and understand the full extent of

---

position in HeLa cells enriched in G1 and G2 mapped to chromosome 11 of the T2T assembly with conventional mapping (E, F) and high-confidence mapping (G, H). (E, G); scale bar 1 Mb, (F, H); scale bar 1 Kb. Color coded panels (F, H) represent high-resolution view of the colored locations indicated in (E, G). MACS2 peaks in (G, H) indicate positions of significantly enriched peaks for CENP-A (MACS2 $P < 0.00001$). Annotation tracks indicate positions of centromeric satellites (see color key in panel (A)) and position of CENP-B boxes. **(I)** Venn diagram depicting the percent overlap of significantly enriched CENP-A peaks identified in data from HeLa cells in G1 and G2 phases (MACS2 $P < 0.00001$, ≥10-fold enrichment). 93% of CENP-A peaks in G1 were retained in G2. Small grey circle represents the relative size (5.1%) of the high-confidence CENP-A peaks depicted in the venn diagram in (J). **(J)** Venn diagram depicting the percent overlap of significantly enriched CENP-A peaks identified in high-confidence CENP-A mapped reads from HeLa cells in G1 and G2 phases (MACS2 $P < 0.00001$, ≥10-fold enrichment). 97% of CENP-A peaks in G2 were precisely maintained from their existing position in G1. **(K)** Enrichment levels of significantly enriched ($P < 0.00001$, ≥10-fold enrichment) high-confidence CENP-A G1 peaks that are either maintained or not maintained into G2. Peaks that were maintained from G1 to G2 had a median enrichment value of 47.65-fold. G1 peaks that were not maintained to G2 had a median enrichment value of 16.26-fold. New G2 peaks that were not present in G1 had a median enrichment value of 19.89-fold.

structure-function relationships between α-satellite DNA and centromere proteins.

The T2T assembly has allowed us to identify a 20-fold increase (from 96 to 1,879) in the number of unique centromeric variants at which CENP-A is bound (Aganezov et al, 2022). Using this expanded set of unique centromeric markers, we find that CENP-A position is precisely maintained through DNA replication by reloading onto the exact same α-satellite sequence and position during DNA replication (Fig 4), which is consistent with our previous findings using the human centromere models in the hg38 assembly. Though ChIP-seq is a population assay that cannot resolve CENP-A position in individual cells, we only find rare examples where CENP-A enrichment on a population level changes dramatically between the G1 and G2 phases, indicating that most of the CENP-A peaks are maintained from G1 to G2 at the population level. This supports our proposed model for the epigenetic maintenance of human centromeres by precise CENP-A reloading and selective CENP-A retention at centromeres, coupled with removal of ectopic CENP-A during DNA replication (Nechemia-Arbely et al, 2019). Collectively, this work demonstrates that plasticity in the position of CENP-A enrichment and location of kinetochore recruitment between human cell lines is accompanied by precise CENP-A maintenance across the cell cycle to preserve unique epigenetic centromere identities among human cells.

# Materials and Methods

## Cell culture

PD-NC4 fibroblasts (Amor et al, 2004) (a kind gift from Ben Black) were immortalized by ectopic retroviral expression of human telomerase (*hTERT*) and oncogenic *KRAS^V12*. Retroviral supernatants were obtained from 293GP cells co-transfected with pVSV-G and pBABE-puro-hTERT or pBABE-hygro-KRAS-V12 constructs (kind gifts from Jerry Shay) using FuGENE HD (Promega) and passed through a 0.45 μm filter. After transduction of PD-NC4 cells in the presence of 5 μg/ml polybrene (Santa Cruz), infected cells were selected with 2 μg/ml puromycin and 100 μg/ml hygromycin, respectively. Cells were maintained in DMEM medium (Gibco) containing 10% FBS (Omega Scientific), 100 U/ml penicillin, 100 U/ml streptomycin, and 2 mM l-glutamine at 37°C in a 5% $CO_2$ atmosphere with 21% oxygen. Cells were maintained and split every 3–4 d according to ATCC recommendations. Single-cell–derived clones were obtained by limited dilution.

## Chromatin extraction and purification

Chromatin was extracted Nuclei from $1 \times 10^8$ nuclei of PD-NC4 cells as previously described (Nechemia-Arbely et al, 2019). CENP-A–bound chromatin was immunoprecipitated using Abcam ab13939 CENP-A antibody coupled to Dynabeads M-270 Epoxy. Chromatin extracts were incubated with antibody-bound beads for 16 h at 4°C. Bound complexes were washed once in buffer A (20 mM HEPES at pH 7.7, 20 mM KCl, 0.4 mM EDTA, and 0.4 mM DTT), once in buffer A with 300 mM KCl, and finally twice in buffer A with 300 mM KCl, 1 mM DTT, and 0.1% Tween 20.

## DNA extraction

After elution of the chromatin from the beads, proteinase K (100 μg/ml) was added, and samples were incubated for 2 h at 55°C. DNA was purified from proteinase K–treated samples using a DNA purification kit following the manufacturer instructions (Zymo Research) and was subsequently analyzed either by running a 2% low melting agarose (APEX) gel or by an Agilent 2100 Bioanalyzer by using the DNA 1000 Kit.

## ChIP-seq library generation and sequencing

ChIP libraries were prepared using NEB Ultra II (Cat # E7103L), following NEBNext protocols with minor modifications. To reduce biases induced by PCR amplification of a repetitive region, libraries were prepared from 80–100 ng of input or ChIP DNA. The DNA was end-repaired and A-tailed, and NEBNext adapters (Cat # E7335S) were ligated. The libraries were PCR-amplified using only five to seven PCR cycles because the starting DNA amount was high. Libraries were run on a 2% agarose gel and size selected for 200–350 bp. Resulting libraries were sequenced using 150 bp, paired-end sequencing on a HiSeq X instrument per the manufacturer's instructions (Illumina).

## Mapping of CENP-A–bound reads

Reads generated from PD-NC4 CENP-A ChIP-seq and from publicly available datasets were assessed for quality using FastQC (https://github.com/s-andrews/FastQC), trimmed with Sickle (https://github.com/najoshi/sickle; v1.33) to remove low-quality 5′ and 3′ end bases, and trimmed with Cutadapt (v.2.10) to remove adapters.

Processed CENP-A ChIP reads were aligned to the CHM13 whole-genome assembly v1.1 using BWA (v0.7.17) with the following parameters: bwa mem -k 50 -c 1000000 [index] [read1.fastq] [read2.fastq] for paired-end data. The resulting SAM files were filtered using SAMtools47 with FLAG score 2308 to prevent multi-mapping of reads. This filter randomly assigns read mapping to more than one location to a single mapping location, preventing mapping biases in highly identical regions. Alignments were normalized with deepTools77 (v.3.3.0) bamCompare with the following parameters: bamCompare -b1 [ChIP.bam] -b2 [Input.bam] –operation ratio –binSize 50 -o [output.bw]. Wiggle tracks for publicly available CUT&RUN data without bulk input were generated with bamCoverage using the following parameters: bamCoverage -b [ChIP.bam] –binSize 50 -o [output.bw].

To visualize high-confidence CENP-A positions, we used mapping scores (MAPQ: 20, or the probability of correctly mapping to another location is 0.01) to identify reads that aligned uniquely to low-frequency repeat variants. We also identified high-confidence CENP-A positions using kmer-assisted mapping (Altemose et al, 2022) and found the results to be similar to MAPQ filtering (Fig S3). MAPQ-filtered reads were normalized to the input using bamCompare with the following normalization parameters: bamCompare -b1 [ChIP.bam] -b2 [Input.bam] –operation ratio –binSize 50 -o [output.bw] –scaleFactors 1:1. The resulting bigWig files were visualized on the UCSC Genome Browser using the CHM13v1.1 assembly.

## ChIP-seq peak calling

Significantly enriched CENP-A peaks were determined with MACS2 (v2.2.7.1) using default parameters, -g $3.03 \times 10^9$ and -q 0.00001. Enrichment scores were determined as the log-transformed normalized value of the ratio between ChIP-seq and background, and those with a score greater than or equal to 10 were included in our study as a high-confidence enrichment set. To identify high-confidence enriched CENP-A peaks that were retained between HeLa G1 and G2 cells, BED files generated from MACS2 peak calling were compared using BEDtools (v2.30.0) intersecting with the following parameters: -a <G1.bed> -b <G2.bed> -u -r -f 0.50.

# Data Availability

PDNC4 ChIP-seq data in this article can be accessed at GEO accession GSE221541. Previously published ChIP-seq and CUT&RUN data that were re-analyzed in this study can be found at the following accession numbers: GSE111381 (HeLa), GSE132193 (RPE-1), GSE60951 (HuRef), and PRJNA559484 (CHM13).

# Supplementary Information

# Acknowledgements

We thank the T2T Consortium for early access to annotation, suggested mapping strategies, and assembly data. We thank Don Cleveland for critical discussion. We thank Glenis Logsdon for technical guidance and discussion and Ben Black for reagents. This work was supported by a grant (R35GM142717) from the NIH to Y Nechemia-Arbely.

## Author Contributions

MA Mahlke: conceptualization, data curation, formal analysis, investigation, methodology, and writing—original draft, review, and editing.
L Lumerman: project administration and writing—review and editing.
P Ly: resources and writing—review and editing.
Y Nechemia-Arbely: conceptualization, funding acquisition, investigation, project administration, and writing—original draft, review, and editing.

## Conflict of Interest Statement

The authors declare that they have no conflict of interest.

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
