## [Reviewer comments · Life Science Alliance]

Epigenetic centromere identity is precisely maintained through DNA replication but is uniquely specified among human cells

Megan Mahlke, Lior Lumerman, Peter Ly and Yael Nechemia-Arbely

DOI: 10.26508/lsa.202201807

Corresponding author(s): Dr. Yael Nechemia-Arbely (University of Pittsburgh)

Review timeline:

Submission Date:	2022-11-07
Editorial Decision:	2022-11-08
Revision Received:	2022-12-16
Editorial Decision:	2022-12-19
Revision Received:	2022-12-21
Accepted:	2022-12-22

Scientific Editor: Eric Sawey

Transaction Report:

Please note that the manuscript was previously reviewed at another journal and the reports were taken into account in the decision-making process at Life Science Alliance.

No Peer Review Process File is available with this article, as the authors have chosen not to make the review process public in this case.

1st Editorial Decision

08 November 2022

Re: Life Science Alliance manuscript #LSA-2022-01807-T

Dr. Yael Nechemia-Arbely

Dept. of Pharmacology and Chemical Biology

University of Pittsburgh

2.32d Hillman Cancer Center

Pittsburgh, PA 15232

Dear Dr. Nechemia-Arbely,

Thank you for submitting your manuscript entitled "Epigenetic centromere identity is precisely maintained through S-phase but is uniquely specified among human cells" to Life Science Alliance. We invite you to submit a revised manuscript addressing the Reviewer comments.

Thank you for this interesting contribution to Life Science Alliance. We are looking forward to receiving your revised manuscript.

Sincerely,

B. MANUSCRIPT ORGANIZATION AND FORMATTING:

2nd Editorial Decision

19 December 2022

RE: Life Science Alliance Manuscript #LSA-2022-01807-TR

Dr. Yael Nechemia-Arbely
University of Pittsburgh
Dept. of Pharmacology and Chemical Biology
2.32d Hillman Cancer Center
Pittsburgh, PA 15232

Dear Dr. Nechemia-Arbely,

Thank you for submitting your revised manuscript entitled "Epigenetic centromere identity is maintained across S-phase but uniquely specified among human cells". We would be happy to publish your paper in Life Science Alliance pending final revisions necessary to meet our formatting guidelines.

- please upload your manuscript text as an editable doc file
 - please upload both your main and supplementary figures as single files
 - please add ORCID ID for corresponding author-you should have received instructions on how to do so
 - please consult our manuscript preparation guidelines <https://www.life-science-alliance.org/manuscript-prep> and make sure your manuscript sections are in the correct order
 - please add a callout for Figure S5A to your main manuscript text
- If you are planning a press release on your work, please inform us immediately to allow informing our production team and scheduling a release date.

A. FINAL FILES:

-- Summary blurb (enter in submission system): A short text summarizing in a single sentence the study (max. 200 characters including spaces). This text is used in conjunction with the titles of papers, hence should be informative and complementary

to the title. It should describe the context and significance of the findings for a general readership; it should be written in the present tense and refer to the work in the third person. Author names should not be mentioned.

B. MANUSCRIPT ORGANIZATION AND FORMATTING:

Sincerely,

3rd Editorial Decision

22 December 2022

RE: Life Science Alliance Manuscript #LSA-2022-01807-TRR

Dr. Yael Nechemia-Arbely
University of Pittsburgh
Dept. of Pharmacology and Chemical Biology
2.32d Hillman Cancer Center

Pittsburgh, PA 15232

Dear Dr. Nechemia-Arbely,

Thank you for submitting your Research Article entitled "Epigenetic centromere identity is maintained across S-phase but uniquely specified among human cells". It is a pleasure to let you know that your manuscript is now accepted for publication in Life Science Alliance. Congratulations on this interesting work.

DISTRIBUTION OF MATERIALS:

Again, congratulations on a very nice paper. I hope you found the review process to be constructive and are pleased with how the manuscript was handled editorially. We look forward to future exciting submissions from your lab.

Sincerely,
